# Unlike Chloroquine, Mefloquine Inhibits SARS-CoV-2 Infection in Physiologically Relevant Cells

**DOI:** 10.3390/v14020374

**Published:** 2022-02-11

**Authors:** Carolina Q. Sacramento, Natalia Fintelman-Rodrigues, Suelen S. G. Dias, Jairo R. Temerozo, Aline de Paula D. Da Silva, Carine S. da Silva, Camilla Blanco, André C. Ferreira, Mayara Mattos, Vinicius C. Soares, Filipe Pereira-Dutra, Milene Dias Miranda, Debora F. Barreto-Vieira, Marcos Alexandre N. da Silva, Suzana S. Santos, Mateo Torres, Otávio Augusto Chaves, Rajith K. R. Rajoli, Alberto Paccanaro, Andrew Owen, Dumith Chequer Bou-Habib, Patrícia T. Bozza, Thiago Moreno L. Souza

**Affiliations:** 1Laboratório de Imunofarmacologia, Oswaldo Cruz Institute, Fundação Oswaldo Cruz (Fiocruz), Rio de Janeiro 21040-360, RJ, Brazil; nataliafintelman@gmail.com (N.F.-R.); suelen.sgdias@gmail.com (S.S.G.D.); aline.paula@hotmail.com.br (A.d.P.D.D.S.); carine.s.silva18@gmail.com (C.S.d.S.); camillablanco.a@gmail.com (C.B.); andre.bio2009@gmail.com (A.C.F.); maymattos03@gmail.com (M.M.); cardosodante42@gmail.com (V.C.S.); filipe.spd@gmail.com (F.P.-D.); otavioaugustochaves@gmail.com (O.A.C.); pbozza@gmail.com (P.T.B.); 2National Institute for Science and Technology on Innovation in Diseases of Neglected Populations (INCT/IDPN), Center for Technological Development in Health (CDTS), Fiocruz, Rio de Janeiro 21040-360, RJ, Brazil; 3National Institute for Science and Technology on Neuroimmunomodulation (INCT/NIM), Oswaldo Cruz Institute, Fiocruz, Rio de Janeiro 21040-360, RJ, Brazil; jairo.jrt@gmail.com (J.R.T.); dumith.chequer@gmail.com (D.C.B.-H.); 4Laboratory on Thymus Research, Oswaldo Cruz Institute, Fiocruz, Rio de Janeiro 21040-360, RJ, Brazil; 5Laboratório de Pesquisas Pré-Clínicas, Departamento de Ciências Biológicas, Universidade Iguaçu, Nova Iguaçu 26260-045, RJ, Brazil; 6Program of Immunology and Inflammation, Federal University of Rio de Janeiro (UFRJ), Rio de Janeiro 21941-617, RJ, Brazil; 7Laboratório de Vírus Respiratório e do Sarampo, Oswaldo Cruz Institute, Fiocruz, Rio de Janeiro 21040-360, RJ, Brazil; milenediasmiranda@gmail.com; 8Laboratório de Morfologia e Morfogênese Viral, Oswaldo Cruz Institute, Fiocruz, Rio de Janeiro 21040-360, RJ, Brazil; barreto@ioc.fiocruz.br (D.F.B.-V.); marquinhosans@gmail.com (M.A.N.d.S.); 9School of Applied Mathematics, Fundação Getulio Vargas, Rio de Janeiro 22250-900, RJ, Brazil; suzana.santos@fgv.br (S.S.S.); mateo.torres@fgv.br (M.T.); alberto.paccanaro@fgv.br (A.P.); 10Centre of Excellence in Long Acting Therapeutics (CELT), Department of Pharmacology and Therapeutics, University of Liverpool, Liverpool L1 8JX, UK; R.K.Rajoli@liverpool.ac.uk (R.K.R.R.); aowen@liverpool.ac.uk (A.O.); 11Department of Computer Science, Royal Holloway, University of London, Egham WC1E 7HU, UK

**Keywords:** SARS-CoV-2, antiviral, COVID-19, antimalarial drug, mefloquine

## Abstract

Despite the development of specific therapies against severe acute respiratory coronavirus 2 (SARS-CoV-2), the continuous investigation of the mechanism of action of clinically approved drugs could provide new information on the druggable steps of virus–host interaction. For example, chloroquine (CQ)/hydroxychloroquine (HCQ) lacks in vitro activity against SARS-CoV-2 in TMPRSS2-expressing cells, such as human pneumocyte cell line Calu-3, and likewise, failed to show clinical benefit in the Solidarity and Recovery clinical trials. Another antimalarial drug, mefloquine, which is not a 4-aminoquinoline like CQ/HCQ, has emerged as a potential anti-SARS-CoV-2 antiviral in vitro and has also been previously repurposed for respiratory diseases. Here, we investigated the anti-SARS-CoV-2 mechanism of action of mefloquine in cells relevant for the physiopathology of COVID-19, such as Calu-3 cells (that recapitulate type II pneumocytes) and monocytes. Molecular pathways modulated by mefloquine were assessed by differential expression analysis, and confirmed by biological assays. A PBPK model was developed to assess mefloquine’s optimal doses for achieving therapeutic concentrations. Mefloquine inhibited SARS-CoV-2 replication in Calu-3, with an EC_50_ of 1.2 µM and EC_90_ of 5.3 µM. It reduced SARS-CoV-2 RNA levels in monocytes and prevented virus-induced enhancement of IL-6 and TNF-α. Mefloquine reduced SARS-CoV-2 entry and synergized with Remdesivir. Mefloquine’s pharmacological parameters are consistent with its plasma exposure in humans and its tissue-to-plasma predicted coefficient points suggesting that mefloquine may accumulate in the lungs. Altogether, our data indicate that mefloquine’s chemical structure could represent an orally available host-acting agent to inhibit virus entry.

## 1. Introduction

Repurposing of clinically approved drugs is considered a rapid way to respond to public health emergencies [1,2]. Indeed, the World Health Organization (WHO) and the University of Oxford launched Solidarity and Recovery clinical trials, respectively, against 2019 coronavirus disease (COVID-19) [3,4] just few months after the severe acute respiratory syndrome coronavirus 2 (SARS-CoV-2) emerged [5]. WHO’s Solidarity trial failed to demonstrate a clinical benefit of chloroquine (CQ)/hydroxychloroquine (HCQ), lopinavir (LPV)/ritonavir (RTV) with or without interferon, and remdesivir (RDV) [6]. LPV/RTV and CQ/HCQ were also not effective in the Recovery trial [7,8]. Conversely, independent clinical investigations of RDV demonstrated its clinical benefit when given early after the onset of illness [9,10,11]. Despite that, RDV access is limited to wealthy countries and its intravenous administration is impractical for early and daily use. The very initial positive clinical results on the specific anti-SARS-CoV-2 RNA polymerase and protease inhibitors, molnupiravir and paxlovid [12,13], respectively, emerged almost two years after the pandemic outbreak. On the other hand, identification of host-acting agents, such as immunomodulatory drugs, has yielded several successful interventions at later stages of the disease, such as steroids and IL-6 antagonists [14].

Although vaccines have been successfully developed at unprecedented speed to combat COVID-19-associated hospitalization and mortality [15], the slow vaccination and antiviral treatments’ global roll out still make COVID-19 a global cause of morbidity and mortality, with over 200,000 deaths/month since its outbreak [16]. Besides, the emergence of immune-escaping SARS-CoV-2 variants has fueled the pandemic, increasing the public health and economic burden [17].

COVID-19 is a virus-triggered systemic disease that can range from a mild to severe inflammatory syndrome, with intense monocyte and lymphocyte death [18]. Thus, the study of the activity of clinically approved compounds against SARS-CoV-2 could still be useful to understand the druggable steps of virus–host interaction and to develop the next generation of antivirals against the variants of concern [19,20,21]. COVID-19 severity is associated with active SARS-CoV-2 replication in type II pneumocytes, cytokine storm, and exacerbation of thrombotic pathways [22,23,24]. Consistent antiviral activity in different cellular systems is not common among the drugs repurposed against COVID-19 [20]. Generally, antiviral compounds identified in Vero cells were not often validated in Calu-3 type II pneumocytes or monocytes. Indeed, poor clinical results of CQ/HCQ [6,8] could have been anticipated by the lack of SARS-CoV-2 susceptibility to these drugs in Calu-3 cells. Therefore, regarding antiviral development, the identification of molecules with chemical structures that surpass the limitations of the first generation of reproposed drugs is worthy.

Very often, antimalarial drugs emerge from compound screening libraries where in vitro activity against SARS-CoV-2 is assessed in unmodified Vero cells [19,20,21]. For instance, CQ/HCQ is a 4-aminoquinoline that blocks virus entry [25] only in cells lacking the expression of TMPRSS2 [26], which does not represent the predominant SARS-CoV-2 entry mechanism, as this receptor is crucial for successful infection in vivo. Mefloquine, which is a 4-quinolinemethanol, shares similar structural areas with CQ/HCQ. However, the changes in the chemical reactivity of the quinoline core results in different physical-chemical properties that are important for mefloquine’s charge and cell permeability in physiological environments, such as the pK and logP values. Mefloquine’s high pK drives lysosomal trapping of the drug in tissues with high lysosome contents, such as in the lungs [20]. In addition, high lung concentrations of mefloquine are driven by its lipophilicity, as it has a high volume of distribution in the lungs and other potential infection sites for SARS-CoV-2 [20]. Although mefloquine has been repurposed for other respiratory diseases (e.g., tuberculosis) [27,28] and is endowed with activity against other highly pathogenic coronaviruses, such as SARS-CoV and Middle East respiratory syndrome coronavirus (MERS-CoV) [29], this drug has been overlooked in clinical trials against COVID-19 (just one active study, NCT04847661, out of more than 5000 registered on clinicaltrials.gov). Despite historical concerns about the safety of mefloquine, in recent years, its safety profile has been positively reviewed [27], such that the United States Food and Drug Administration (FDA) modified mefloquine’s safety from class C to B during pregnancy [30].

Robust pre-clinical evaluation that allows the identification of the drug’s mechanisms of action also provides useful information for further development and evidence-based decision-making for compounds to progress to clinical trials. Here, we showed that SARS-CoV-2 is susceptible to mefloquine in type II pneumocytes and monocytes. Our results indicate that mefloquine is a host-acting agent targeting the early stages of the virus’ life cycle. Since our previous work indicated that antiviral concentrations of mefloquine may be achievable at approved doses of the drug [20], we developed a physiologically based pharmacokinetic model (PBPK) to propose optimal doses for achieving therapeutic concentrations.

## 2. Materials and Methods

### 2.1. Reagents

Mefloquine (MQ) and chloroquine (CQ) were received as donations from Instituto de Tecnologia de Fármacos (Farmanguinhos, Fiocruz). Remdesivir (RDV) was purchased from Selleckhem. Drugs were dissolved in 100% dimethylsulfoxide (DMSO) and diluted at least 10^4^-fold in culture medium before each assay. The materials for cell culture were purchased from Thermo Scientific Life Sciences, unless otherwise mentioned. Kits for ELISA assays were purchased from R&D Bioscience.

### 2.2. Cells and Virus

African green monkey kidney (Vero, subtype E6) and human lung epithelial (Calu-3) cells were cultured in DMEM high glucose, complemented with 10% fetal bovine serum (FBS; HyClone, Logan, UT, USA), 100 U/mL penicillin, and 100 μg/mL streptomycin (P/S). Primary monocytes were obtained through 3-h adherence to plastic from 2.0 × 10^6^ peripheral blood mononuclear cells (PBMCs; obtained from healthy donors by density gradient centrifugation, Ficoll-Paque, GE Healthcare, Chicago, IL, USA) onto 48-well plates (NalgeNunc, Rochester, NY, USA) in RPMI-1640 without serum. Non-adherent cells were washed, and the remaining monocytes were maintained in DMEM high glucose with 5% human serum (HS; Millipore, Burlington, MA, USA) and P/S. The purity of human monocytes was above 95%, as determined by flow cytometric analysis (FACScan; Becton Dickinson, Franklin Lakes, NJ, USA) using anti-CD3 (BD Biosciences, Franklin Lakes, NJ, USA) and anti-CD16 (Southern Biotech, Birmingham, AL, USA) monoclonal antibodies. Cells were kept at 5% CO_2_ and 37 °C.

The SARS-CoV-2 D614G (GenBank #MT710714) and Gamma strains (#EPI_ISL_1060902) were grown in Vero E6 cells at a multiplicity of infection (MOI) of 0.01. Viruses’ preparations were handled at biosafety level 3 (BSL3) and titers were determined as plaque-forming units (PFU) per milliliter (mL). Virus stocks were kept in −80 °C ultralow freezers.

### 2.3. Cytotoxicity Assay

Monolayers of 2 × 10^4^ (Vero E6) or 2 × 10^5^ (Calu-3) cells in 96-well plates were treated with various concentrations of the tested drugs for 72 h at 5% CO_2_ and 37 °C. Culture medium was removed, cells were washed with PBS, and incubated for 1 h at 37 °C with a 1.25% glutaraldehyde and 0.6% methylene blue solution diluted in Hank’s balanced salt solution (HBSS). Cells were washed, dried at room temperature (RT), and incubated with the elution solution (50% ethanol, 49% PBS, and 1% acetic acid) for 1 h at RT. The elution solution was transferred to a new 96-well plate and, after centrifugation at 12,000× *g* for 3 min, the plate was read at 570 nm [31].

### 2.4. Yield Reduction Assay and Virus Titration

Vero E6 (2 × 10^4^/well) and Calu-3 (2 × 10^5^/well) in 96-well plates, and primary monocytes (10^5^/well) in 48-well plates were infected with SARS-CoV-2 at MOI of 0.01 (Vero E6 and monocytes) or 0.1 (Calu-3). After a 1-h incubation period, inoculum was removed and cells were treated with different concentrations of the tested drugs in complete medium. After 24 (Vero E6 and monocytes) or 48 h (Calu-3), supernatants were collected, and viruses were quantified by titration or real-time RT-PCR.

### 2.5. Virus Titration

Vero E6 in 96-well plates (2 × 10^4^ cells/well) were infected with serial dilutions of the yield reduction assays’ supernatants containing SARS-CoV-2 for 1 h at 37 °C. Semi-solid high-glucose DMEM medium containing 2% FSB and 2.4% carboxymethylcellulose (CMC) was added and cultures were incubated for 3 days at 37 °C. Cells were fixed with 3.7% formaline for 2 h at RT. The cell monolayer was stained with 0.04% solution of crystal violet in 20% ethanol for 1 h. The virus titers were determined by PFU/mL.

### 2.6. Quantification of Viral RNA Levels

The total viral RNA from culture supernatants and/or monolayers was extracted using QIAamp Viral RNA (Qiagen^®^), according to the manufacturer’s instructions. Quantitative RT-PCR was performed using a GoTaq^®^ Probe qPCR and RT-qPCR Systems (Promega, Madison, WS, USA) in a StepOne™ Real-Time PCR System (Thermo Fisher Scientific, Waltham, MA, USA). Primers, probes, and cycling conditions recommended by the Centers for Disease Control and Prevention (CDC) protocol were used to detect SARS-CoV-2 [32]. The standard curve method was employed for virus quantification. For reference to the cell amounts used, the housekeeping gene RNAse P was amplified.

### 2.7. Adsorption Inhibition Assays

To evaluate the effects of mefloquine on SARS-CoV-2 attachment, Calu-3 in 96-well plates (2 × 10^5^/well) were infected with MOI of 0.1 for 1 h at 37 °C. Two different approaches were used: (i) SARS-CoV-2 was pre-incubated with 1 μM of mefloquine for 1 h at 37 ºC and then used to infect Calu-3 for an additional hour; or (ii) Calu-3 were pre-treated with 1 μM of mefloquine for 1 h at 37 °C and then infected with SARS-CoV-2. After 24 h, supernatants were collected for virus titration through PFU/mL.

### 2.8. Measurement of Inflammatory Mediators and Cell Death Marker

The levels of IL-6, TNF-α, and extracellular lactate dehydrogenase (LDH) were quantified in the supernatants of SARS-CoV-2-infected monocytes using commercially available ELISA kits (for cytokines) or Doles^®^ kit (Promega, for LDH, Madison, WS, USA), according to the manufacturer’s instructions, as we described elsewhere [33].

### 2.9. Transmission Electron Microscopy

Vero E6 cells (2 × 10^5^ cells in 25 cm^2^ culture flasks) were infected with SARS-CoV-2 at an MOI of 1 for 1 h at 37 °C. Inoculum was removed and cells were treated with 5 μM of mefloquine. After 4 h, cells were washed with PBS and fixed in 2.5% glutaraldehyde in sodium cacodilate buffer (0.2 M, pH 7.2), post-fixed in 1% buffered osmium tetroxide, dehydrated in acetone, embedded in epoxy resin, and polymerized at 60 °C over the course of 3 days [34,35]. Ultrathin sections (50–70 nm) were obtained from the resin blocks. The sections were picked up using copper grids, stained with uranyl acetate and lead citrate [36], and observed using a Hitachi HT 7800 transmission electron microscope.

### 2.10. Differential Expression Analysis

We performed three different analyses of the differential expression. The first aimed to identify pathways targeted by mefloquine. We used data from the Connectivity Map (CMAP) [37,38], a database containing the gene expression profiles of cells treated with thousands of compounds. Among the different cell lines available in CMAP, we chose data from A549 cells, which is the one most similar to the Calu-3 used in the biological experiments. The set of genes comprising the mefloquine expression signature was defined using modified z-scores obtained when comparing mefloquine-treated cells with controls across 3 replicates. The data is available from GEO (https://www.ncbi.nlm.nih.gov/geo/, accessed on 5 June 2021) [39], accession number GSE92742. To identify pathways targeted by mefloquine, we applied GSEA [40] using KEGG (Kyoto Encyclopedia of Genes and Genomes) pathways [41]. GSEA computes an enrichment score that measures how much a pathway is enriched in differentially expressed genes. Pathways with FDR q-value < 0.05 and a positive enrichment score were considered to be significantly upregulated.

The second analysis aimed to identify pathways affected by SARS-CoV-2 infection. We downloaded gene expression data from GEO (accession number GSE148729) [42]. This consists of RNAseq raw reads obtained from two replicates of SARS-CoV-2-infected Calu-3 cells (after 4 h of infection) and controls. Differentially expressed genes comprising the SARS-CoV-2 infection expression signature were identified using the R package edgeR [43]. We then applied GSEA against the KEGG database. Pathways with an FDR q-value < 0.05 and negative enrichment score were considered to be significantly downregulated.

The third analysis aimed to evaluate whether mefloquine is likely to have therapeutic effects against COVID-19 based on its gene expression profiles. Here, we used the CMAP pipeline [37,38], comparing the mefloquine and SARS-CoV-2 infection expression signatures. The pipeline computes the connectivity score (τ), a standardized measure ranging from −100 to 100. This score is obtained by comparing the enrichment score of a drug to all others in a reference database. A negative τ indicates that the drug and disease have opposite gene expression signatures, suggesting that the drug could revert the effects of the disease [44]. A connectivity score below −90 was considered statistically significant.

### 2.11. Computational Calculations of the Structural and Physical-Chemical Properties

The chemical structures of CQ, HCQ, and mefloquine were built and minimized in terms of energy by density functional theory (DFT) in aqueous media at pH 7.4 (physiological) and 4.0 (endosomal) by Spartan’18 software (https://www.wavefun.com/-Wavefunction, Inc., Irvine, CA, USA). For each compound, the physical-chemical and structural properties, i.e., polarizability, hydrogen donor count, hydrogen acceptor count, area, volume, polar surface area, ovality, the highest occupied molecular orbital (HOMO and HOMO-1), the lowest unoccupied molecular orbital (LUMO and LUMO + 1) and water/octanol partition (logP), were obtained using the DFT method.

### 2.12. PBPK Model

A whole-body PBPK model was developed in Julia programming language v1.5.3 (in Juno v0.8.4) [45] using the packages DataFrames v0.21.7, Differential Equations v6.15.0 and Distributions v0.23.8. The PBPK model consists of various compartments representing the various organs and tissues of the human body. Mefloquine’s physicochemical and drug-specific parameters used in the construction of the PBPK model are shown (Appendix A). The data from this model was exclusively computer generated; therefore, no ethics approval was required for this study.

### 2.13. Model Development

A virtual healthy population of 100 individuals (50% female) between the ages of 18 and 75 years was used in this study and the characteristics, such as the weight and height of the individual, were randomly generated (using an inbuilt programming functionality) from a population with a mean and standard deviation such that every individual was unique [46]. Various anthropometric equations were used to compute individual organ and tissue weights/volumes [47] and blood flow rates [48]. The tissue to plasma ratios were computed with mefloquine’s physicochemical characteristics, such as pKa, logP, and fraction unbound, using equations from a previous publication [49]. For oral drugs, the small intestine was divided into seven compartments and a simultaneous compartmental absorption and transit (CAT) model was used to simulate the physiological absorption process [50]. The drug disposition was defined using various flow equations [51]. The PBPK model has a few limitations for this study: (1) the entire administered drug is in solution and available for absorption, (2) no reabsorption from the large intestine, (3) drug distribution is bloodflow limited, and (4) an instant and uniform drug distribution across a tissue/organ.

### 2.14. Model Validation

Mefloquine’s model was validated against single doses of 250 and 1500 mg and also against multiple doses of 250 mg across various clinical studies. The observed data points were digitized using Web-Plot-Digitiser^®^. The mefloquine PBPK model was assumed to be validated if the difference between the simulated and observed data points computed using the absolute average fold-error was less than two and the mean simulated pharmacokinetic parameters, such as AUC, C_max_ and C_trough_, were less than two-fold from the observed values.

### 2.15. Model Simulations

The EC_90_ value from the experimental settings was 1200 and 2005 ng/mL in the Vero E6 and Calu-3 cell lines, respectively. However, other literature has indicated different EC_90_ values in Vero E6 cells; therefore, an average across the available studies (2311 ng/mL, Appendix A) was used and doses were optimized based on the higher EC_90_ value (Vero E6 compared to Calu-3). Optimal doses were predicted such that mefloquine’s trough concentrations were over the average EC_90_ value in Vero E6 cells (2311 ng/mL) from day 3 to day 7 for at least 90% of the simulated population.

### 2.16. Statistical Analysis

The assays were performed blinded by one professional, codified, and then read by another professional. All experiments were carried out at least three independent times, including a minimum of two technical replicates in each assay. The dose–response curves used to calculate the EC_50_ and CC_50_ values were generated by a variable slope plot from Prism GraphPad software 8.0. The equations used to fit the best curve were generated based on R^2^ values ≥ 0.9. Student’s T-test was used to access statistically significant *p* values < 0.05. The statistical analyses specific to each software program used in the bioinformatics analysis are described above.

### 2.17. Ethics Statement

Experimental procedures involving human cells from healthy donors were performed with samples obtained under the approval of the Institutional Review Board (IRB) of the Oswaldo Cruz Institute/Fiocruz (Rio de Janeiro, RJ, Brazil) under the number 397-07.

## 3. Results

### 3.1. Mefloquine Inhibits SARS-CoV-2 Replication in Calu-3 Cells

Although mefloquine has emerged as a potential anti-SARS-CoV-2 drug through in vitro screening using Vero cells [19,21,52], its antiviral effect in physiologically relevant cellular systems on COVID-19 have not been studied. Calu-3 and Vero E6 were infected with SARS-CoV-2 (MOI of 0.1 and 0.01, respectively). After 48 (Calu-3) or 24 h (Vero E6), virus replication was assessed in the culture supernatants using the plaque assay. Mefloquine inhibited SARS-CoV-2 replication in Calu-3 and Vero E6 in a dose-dependent manner, with EC_50_ values of 1.9 and 0.6 μM, respectively (Table 1 and Figure 1A–D). For comparison, CQ presented an EC_50_ of 1.1 μM in Vero E6 (Table 1 and Figure 1C,D), but it was inactive in Calu-3 [53]. In contrast, mefloquine showed efficient inhibition of virus replication, with optimal inhibitory concentrations ranging from 3.2 to 5.3 μM for Vero E6 and Calu-3, respectively (Table 1 and Figure 1A–D). Mefloquine’s ability to be a donor/acceptor of hydrogen bonds retaining a lipophilic condition in a physiological environment could make its chemical structure more promising than 4-aminoquinolines, such as CQ (Appendix A). RDV, used as a positive control, also displayed suppressive EC_90_ values from 0.3 to 3.3 μM for Calu-3 and Vero E6, respectively (Table 1 and Figure 1A–D). The selectivity index (SI), which represents the margin of in vitro safety based on CC_50_ and EC_50_ values, highlights that mefloquine exhibits its antiviral activity without cytotoxicity. Moreover, we confirmed that the SARS-CoV-2 variant of concern, the gamma strain, is slightly more susceptible to mefloquine in Calu-3 cells, at the EC_90_ level, compared to its predecessor strain (D614G; Figure 1E).

SARS-CoV-2 can infect immune cells, such as monocytes [54], that are key components of the host antiviral response and produce cytokine storm-related mediators in patients with severe COVID-19 [55]. Although human primary monocytes do not produce infectious virus particles, they are susceptible to infection and produce viral RNA [56], leading to massive cell death and leukopenia [57]. Mefloquine reduced viral RNA levels in SARS-CoV-2-infected monocytes by 60%, and CQ and RDV (Figure 2A). At 10 μM, mefloquine reduce virus-induced monocyte death (Figure 2B). Along with cell death, SARS-CoV-2 induced an enhancement of IL-6 and TNF-α levels (in a lower magnitude than LPS, used as a positive control), but mefloquine at 10 μM could reduce the augmentation of these cytokines in a donor-dependent manner of virus-induced IL-6 and TNF-α levels (Figure 2C–F).

### 3.2. Mefloquine Targets Endosome-Related Pathways to Antagonize SARS-CoV-2-Induced Host-Cell Expression

Antimalarial drugs are endowed with the ability to disrupt virus endocytosis [52,58]. To obtain insights into whether mefloquine would also affect endosome-related pathways, we identified differentially expressed pathways in cell lines treated with mefloquine. Out of 162 KEGG pathways, we found 10 that showed significant positive enrichment scores (FDR q-value < 0.05), meaning that 10 cellular pathways are upregulated by mefloquine (Appendix A). Among them, six are indeed involved in the endocytosis process (Figure 3).

To verify whether mefloquine has potential therapeutic effects against COVID-19, we followed an approach first proposed by Sirota et al. [44] in which we tested whether the effects of mefloquine on these pathways and global gene expression are the opposite to the effects induced by SARS-CoV-2 infection. First, we identified differentially expressed pathways in Calu-3 cells infected by SARS-CoV-2. We found that two out of the six endocytosis-related pathways upregulated by mefloquine are downregulated by the infection (Appendix A), suggesting that mefloquine can oppose the effect caused by SARS-CoV-2 infection. Then, we ran the CMAP pipeline [37,38] to test whether the global mefloquine gene expression signature is the opposite to SARS-CoV-2 infection. We found a significant CMAP connectivity score (τ = −90.45), reinforcing the evidence for the potential therapeutic effects of mefloquine.

To experimentally confirm that mefloquine impairs SARS-CoV-2 endocytosis-mediated entry, either Calu-3 cells or the virus were pre-treated with mefloquine. The pre-treatment was performed to limit the steps of the virus’ replicative cycle so we could evaluate this drug’s effect on the entry process. Subsequently, Calu-3 cells were infected with SARS-CoV-2, and virus production was assessed 24 h after infection. In both conditions, mefloquine reduced SARS-CoV-2 replication by 3-log_10_ (Figure 4A). Within 4 h after infection, SARS-CoV-2 entered the host cells (Figure 4B,C) and, in fact, mefloquine reduced the number of viral particles in the endosomes (Figure 4B–D). Our results indicate that mefloquine is an inhibitor of SARS-CoV-2 entry.

Since mefloquine inhibits SARS-CoV-2 entry, it could be expected to synergize with drugs acting on other steps of viral replication. As a proof of concept, we combined mefloquine with remdesivir. Remdesivir’s efficiency was improved by 200-fold in the presence of suboptimal concentrations of mefloquine to inhibit 25 to 50% of virus replication (Figure 5). These findings indicate that mefloquine could be used as an accessory drug to improve the activity of compounds acting on other steps of the virus life cycle.

### 3.3. Mefloquine PBPK Modeling and Dose Simulations

Mefloquine’s PBPK model was validated against available clinical data and the AAFE of the simulated plots was between 0.96 and 1.21. Appendix A show the comparison between the observed and simulated data for various dosing strategies. A scaling factor for intrinsic clearance and tissue plasma ratios of 2 and 0.38, respectively, were applied to validate the model. Furthermore, the bioavailability increased by 10% from day 7 onwards to match the clinical data for multiple dosing studies as mefloquine seems to exhibit higher bioavailability for successive doses.

Various dose settings were assessed and the best two doses, i.e., 450 mg TID and 350 mg QID, for 3 days are shown in Figure 6A,B, such that the plasma concentrations of mefloquine are over the target EC_90_ value in 90% of the simulated population from day 3 to 7. However, much higher doses of mefloquine (1400 mg TID or 1100 mg QID on day 1 only, data not shown) would be required if the plasma concentrations had to reach the target EC_90_ on day 1 and may not be practically possible to administer in individuals. Appendix A show the median C_trough_ on day 1, 3, and 7 and the percentage of the population over the target EC_90_ value of 2311 ng/mL (or 8.3 µM) for the TID and QID settings across various doses.

Although the plasma exposure to reach the EC_90_ would require a new therapeutic dosage and regimen, the EC_50_ (756 ng/mL or 2 µM) and EC_25_ (378 ng/mL or 1 µM) are achievable after a single dose of mefloquine at 1500 mg for over 7 days (Appendix A) and has been demonstrated to enhance remdesivir’s activity.

## 4. Discussion

The clinical evolution of mild to severe COVID-19 is characterized as a multi-stage disease [18], in which acute viral replication is overwhelmed by an intense proinflammatory response along with coagulopathy [22,59,60]. Antivirals are expected to be used early after COVID-19 diagnosis to reduce hospitalization, as evidenced by the clinical trials on molnupiravir and paxlovid [12,13]. These new orally available drugs are apparently more effective in patients than intravenous administration of RDV. RDV showed clinical benefits in Wang et al.’s and Beigel et al.’s studies [9,11] but failed to do so in other works [6]. Orally available drugs endowed with the ability to reduce SARS-CoV-2 levels could reduce the chain of transmission, hospitalization, multi-organ impairment, and mortality. Mefloquine is a lipophilic compound able to cross complex biological barriers that could be potentially interesting against pulmonary and extra-pulmonary manifestations of COVID-19. Here, we showed that this drug inhibited SARS-CoV-2 replication by blocking virus entry through a multi-modulation of cellular pathways, decreasing inflammatory activity and enhancing RDV’s activity with concentrations close to those achievable with approved doses.

Host-directed broad-spectrum antimicrobial drugs have been attempted to be used against COVID-19, such as CQ/HCQ and nitazoxanide [6,61]. Initial enthusiasm towards CQ/HCQ was not sustained scientifically [6,8]. The clinical failure of CQ/HCQ to enhance the recovery of COVID-19 patients could have been anticipated by in vitro experimentation and assessment of its pharmacokinetic performance [6,53]. CQ/HCQ lacks antiviral activity on SARS-CoV-2-infected Calu-3 cells [53], which express TMPRSS2. While the effect of 4-aminoquinolines on endosomal pH could have utility for viruses that depend on this organelle [52,58], the lack of activity in SARS-CoV-2 is now clear. On the other hand, it has been shown that CQ’s anti-SARS-CoV-2 activity may be enhanced by zinc [62]. Similarly, mefloquine’s quinoline moiety may bind cations, based on the frontier molecular orbitals (FMOs) and Pearson acid-base concept (hard and soft acids and bases, HSAB) [63]. However, mefloquine’s nitrogen at the quinoline moiety has a pK lower than 3.0 (Appendix A), being more favorable for the binding of H^+^ in acid endosomes than zinc, for instance.

Amongst antimalarial agents, mefloquine displays good pharmacological properties to be repurposed against COVID-19. Unlike 4-aminoquinolines, mefloquine inhibited SARS-CoV-2 entry in Calu-3 cells, a model for type II pneumocytes, and monocytes, the most affected cells in the respiratory tract of patients that died due to COVID-19 [22,55]. Mefloquine’s tissue-to-plasma predicted coefficient indicates that this drug may accumulate in the lungs [20], predominantly via high lipophilicity and pK-like 4-aminoquinolines. Indeed, mefloquine has been repurposed for tuberculosis, another respiratory tract disease [28], and is currently under investigation as a prophylaxis agent against COVID-19 (ClinicalTrials.gov Identifier: NCT04847661). Importantly, mefloquine’s in vitro pharmacological parameters described here are at least 5-fold better than when compared to tuberculosis, which requires a minimal inhibitory concentration of 20 µM [64].

Due to its high lipophilicity, mefloquine is well distributed throughout anatomical compartments and has a long half-life, measured in weeks [65], meaning that a single or few oral doses could be combined with other antivirals. Although there might be caveats in the conversion of in vitro doses to in vivo, mefloquine’s C_max_ of 3279 ng/mL (equivalent to 8.67 µM) [66] and our in vitro pharmacological parameters ranging from 0.6 to 5.4 µM suggest that the antiviral activity described here may be physiologically achievable. For comparison, CQ’s C_max_ is 870 ng/mL (2.73 µM) [67]. Because the EC_90_ values for CQ in Vero E6 cells are higher than its C_max_, the perspective of its clinical benefit for COVID-19 patients is unfeasible. On the other hand, human plasmatic exposure to mefloquine’s EC_90_ could be achievable under new therapeutic regimens.

Mefloquine does exhibit high binding to plasma proteins (>98%) and this adds a degree of uncertainty regarding its in vivo activity. However, all in vitro experiments reported here were performed in the presence of fetal bovine or human serum, which means that the pharmacological parameters already consider a high level of protein binding [68]. In fact, serum proteins were included in the in vitro activity assays, and the importance of protein binding depends on multiple drug- and target-specific principles [68]. Moreover, even under the current therapeutic approach, suboptimal doses of mefloquine could enhance the anti-SARS-CoV-2 activity of drugs acting on different steps of the virus life cycle, as demonstrated here for RDV, but also could likely be applicable to other RNA polymerase inhibitors, such as molnupinavir and favipiravir. In addition, the short half-life and the accumulation of pro-drug nucleotides (proTides) in the liver may limit the adequate bioavailability of RDV in the respiratory tract [69]. Consequently, the combination of mefloquine and RDV could be beneficial. Indeed, mefloquine alters adenine and purine metabolism [70], which could explain its beneficial effect when combined with RDV.

Altogether, our results indicate that the chemical structure of mefloquine is promising for developing host-acting drugs that affect SARS-CoV-2 entry in a multi-model way.

## 5. Conclusions

Our investigation indicates that mefloquine’s chemical structure may be promising for the development of host-acting antiviral drugs, because: (i) it reduces SARS-CoV-2 entry in Calu-3 cells and virus-induced enhancement of IL-6 and TNF-α in monocytes; (ii) it may synergize with drugs acting on other steps of the virus’ replicative cycle, such as RDV; and (iii) new therapeutic regimens may be achievable within physiologically relevant plasma exposures to inhibit SARS-CoV-2 in vivo.

## Figures and Tables

**Figure 1 viruses-14-00374-f001:**
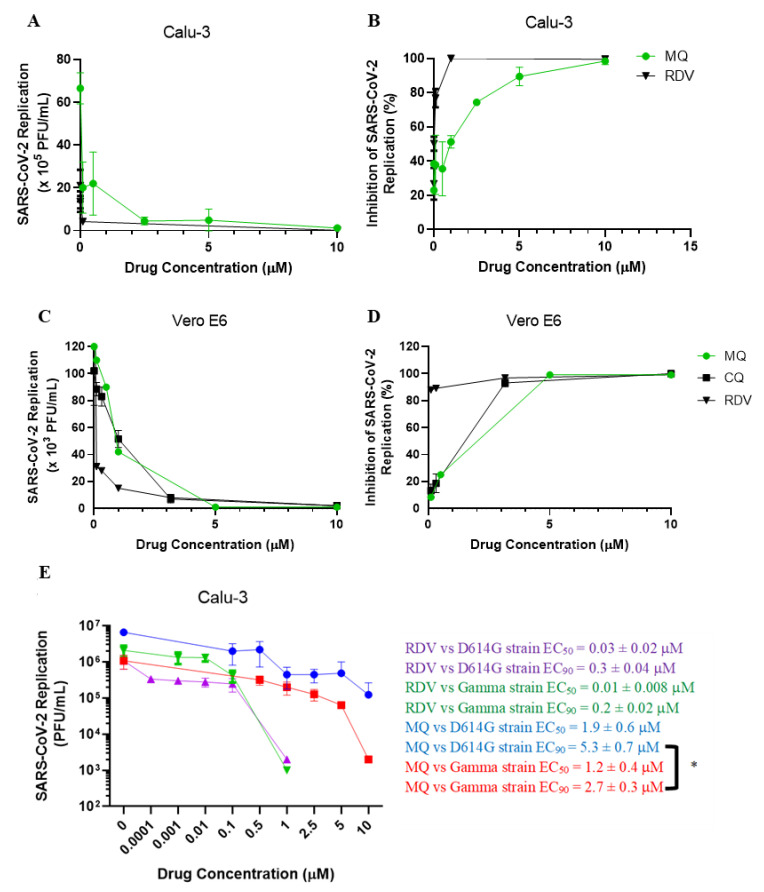
The antiviral effect of mefloquine against SARS-CoV-2. Calu-3 (**A**,**B**,**E**) or Vero E6 (**C**,**D**) were infected with SARS-CoV-2 D614G (**A**–**D**) or gamma strain € at MOI 0.01 (Vero) or 0.1 (Calu-3), for 1 h at 37 °C. Inoculum was removed and cells were incubated with fresh DMEM containing the indicated concentrations of mefloquine (MQ), chloroquine (CQ), or remdesivir (RDV). Virus titers were measured by PFU/mL in the culture supernatants at 24 h post infection (hpi) for Vero (**C**,**D**) or 48 hpi for Calu-3 (**A**,**B**,**E**) cells. Results are displayed as virus titers (**A**,**C**,**E**) or percentage of inhibition (**B**,**D**). The data represent the means ± SEM of three independent experiments. * *p* < 0.05.

**Figure 2 viruses-14-00374-f002:**
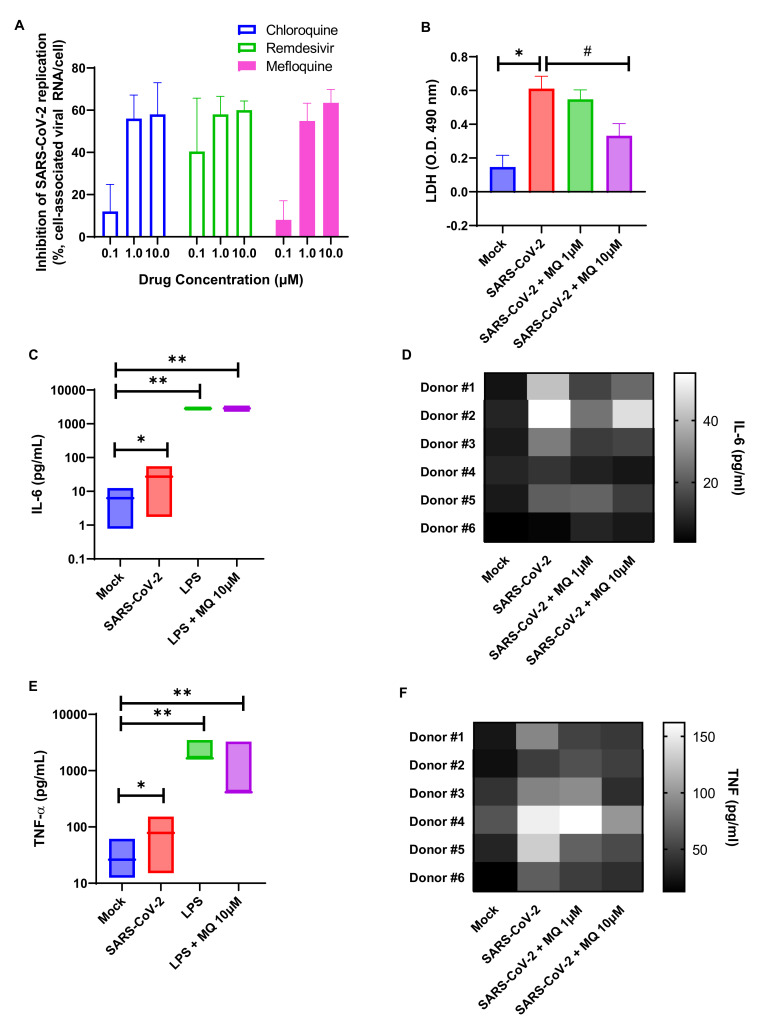
Mefloquine reduces the viral load and viral-induced enhancement of IL-6 and TNF-α in SARS-CoV-2-infected monocytes. Primary human monocytes isolated from health donors were infected with SARS-CoV-2 at an MOI of 0.01 for 1 h at 37 °C. Inoculum was removed and cells were treated with the indicated concentrations of mefloquine (MQ), chloroquine (CQ), or remdesivir (RDV). Virus replication (**A**) and LDH (**B**), IL-6 (**C**,**D**), or TNF-α (**E**,**F**) levels were assessed in the culture supernatants at 24 h post-infection. The data represent the means ± SEM of monocytes from at least 5 healthy donors. Ns: not significant, * *p* < 0.05, ^#^
*p* < 0.05 and ** *p* < 0.01.

**Figure 3 viruses-14-00374-f003:**
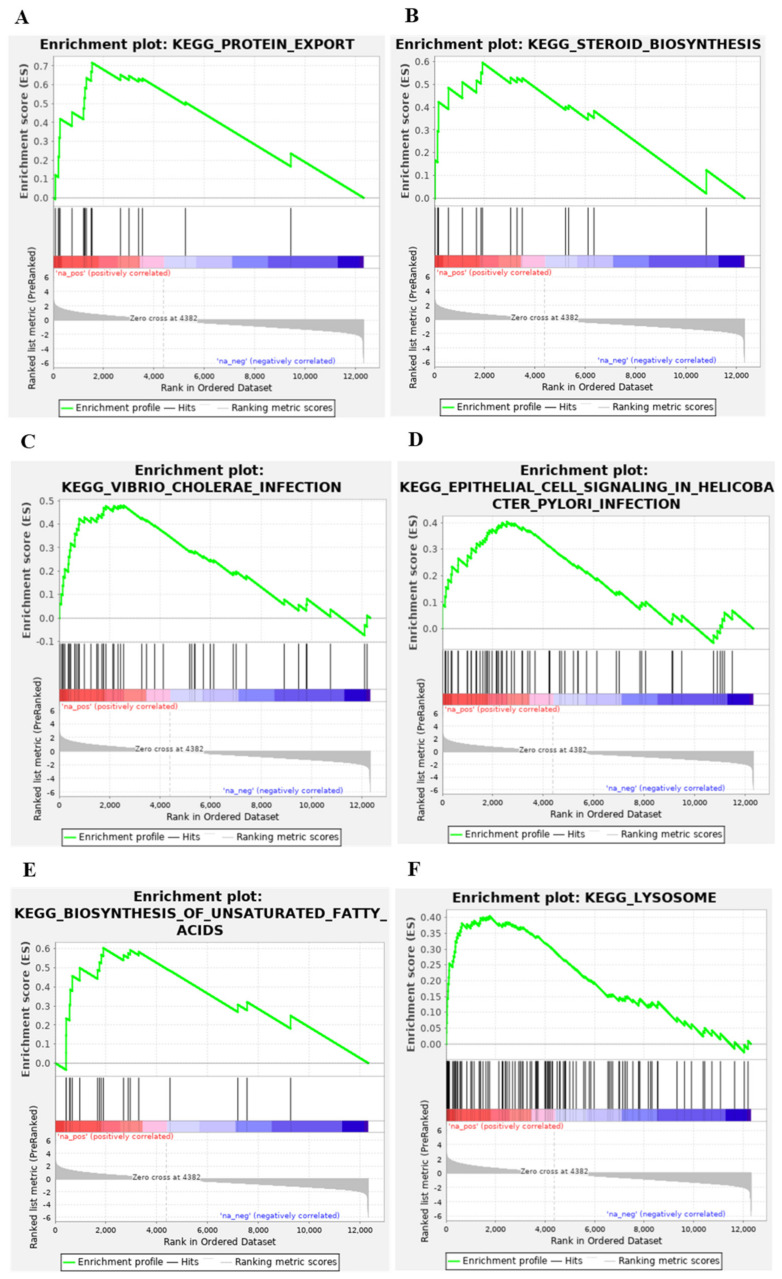
GSEA enrichment plots of the six endocytosis-related pathways upregulated by mefloquine. We ran GSEA on the mefloquine gene expression signature obtained from CMAP to identify KEGG pathways enriched in differentially expressed genes. The six endocytosis-related pathways with a significant enrichment score (FDR < 0.05) are shown here. Subplots contain an identical colored bar, representing the complete gene list ordered by changes in the expression levels induced by mefloquine, with red indicating the highest z-scores (upregulated genes), and blue the lowest ones (downregulated genes). Each subplot corresponds to a pathway and black vertical lines indicate the position of its genes in the ordered list. Note that for all six pathways, these lines tend to be located towards the top of list (upregulated genes). The GSEA enrichment score is computed iteratively across the ordered list, as shown in green. The final GSEA enrichment score of each pathway (reported on Appendix A) is the highest deviation from zero. The bottom portion of the plot shows the z-score (*y*-axis) at each position of the list (*x*-axis).

**Figure 4 viruses-14-00374-f004:**
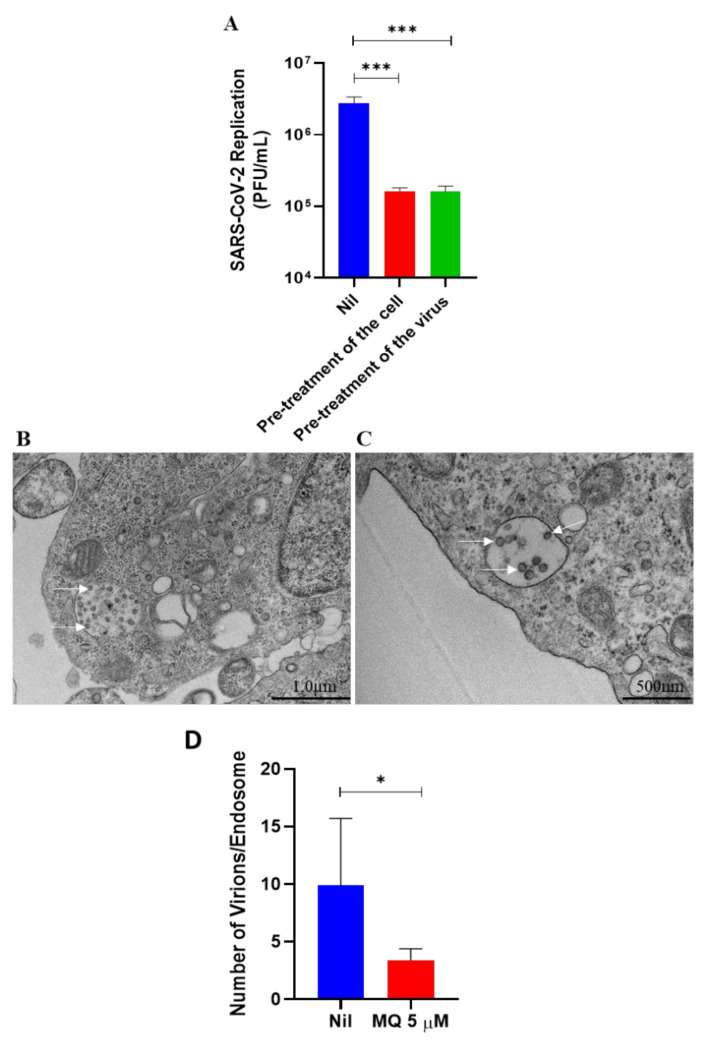
Effects of mefloquine on SARS-CoV-2 endocytosis-mediated entry. (**A**) To initially evaluate mefloquine’s effect on virus entry, Calu-3 cells or SARS-CoV-2 virus particles were pre-incubated with 1 µM of mefloquine for 1 h at 37 °C and then infected at an MOI of 0.1. After 24 hpi, culture supernatants were harvested, and SARS-CoV-2 replication was measured using the plaque assay. Results are displayed as virus titers (PFU/mL). (**B**,**C**) Representative images (from 4 independent experiments) of ultrastructural analysis by transmission electron microscopy of Vero E6 cell infected with an MOI of 1 of SARS-CoV-2 (**B**) and treated with 5 µM of mefloquine for 4 h (**C**). Cell endosomes with spherical SARS-CoV-2 virus particles (white arrows). (**D**) SARS-CoV-2 virus particles were counted inside the endosomes of Vero E6 cells treated with mefloquine or not (nil). * *p* < 0.05 and *** *p* < 0.01.

**Figure 5 viruses-14-00374-f005:**
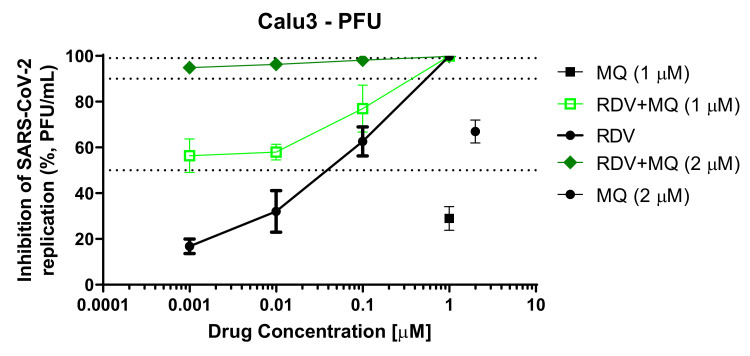
Mefloquine enhances the anti-SARS-CoV-2 activity of remdesivir. Calu-3 cells were infected with SARS-CoV-2 at an MOI 0.1 for 1 h at 37 °C. Inoculum was removed and cells were treated with 1 or 2 µM of mefloquine combined with the indicated concentrations of remdesivir (RDV). Virus titers were measured by PFU/mL in the culture supernatants after 48 hpi. Results are displayed as the percentage of inhibition.

**Figure 6 viruses-14-00374-f006:**
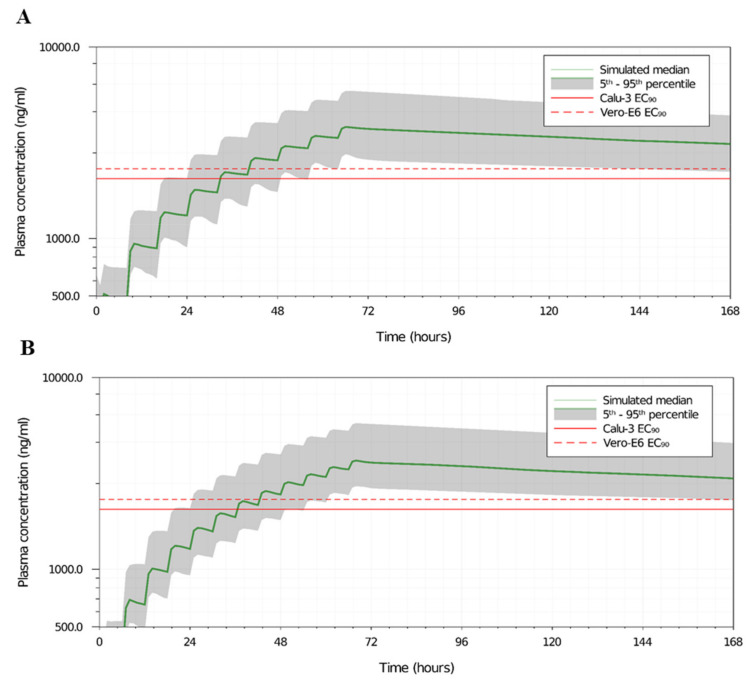
Mefloquine’s PBPK model predicted the plasma concentration for different dosages. Predicted mefloquine plasma concentration for 450 mg TID (**A**) or 350 mg QID (**B**) doses for 3 days. The dotted and the solid lines represent the EC_90_ values of mefloquine for SARS-CoV-2 in Vero E6 and Calu-3 cell types, respectively.

**Table 1 viruses-14-00374-t001:** Chemical structures and pharmacological parameters of mefloquine, chloroquine, and remdesivir in SARS-CoV-2-infected cells.

	Mefloquine (µM) 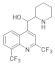	Chloroquine (µM) 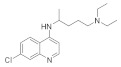	Remdesivir (µM) 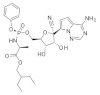
Cell Types	CC_50_	EC_50_	EC_90_	SI	CC_50_	EC_50_	EC_90_	SI	CC_50_	EC_50_	EC_90_	SI
Calu-3	19.9 ± 0.06	1.9 ± 0.6	5.3 ± 0.7	11	ND	>10	>10	ND	480 ± 20	0.03 ± 0.02	0.3 ± 0.04	1.6 × 10^4^
Vero E6	9.2 ± 0.3	0.6 ± 0.03	3.2 ± 0.3	16	259 ± 5	1.1 ± 0.07	3.5 ± 0.8	235	512 ± 30	1.4 ± 0.3	3.3 ± 0.2	366

SI: selectivity Index (calculated based on the ratio of the CC_50_ and EC_50_values); ND: not determined.

## Data Availability

The data presented in this study are available in the article or Appendix A.

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
