# Peer review of "Unlike Chloroquine, Mefloquine Inhibits SARS-CoV-2 Infection in Physiologically Relevant Cells"

_viruses, 2022, doi:10.3390/v14020374_

Round 1

Reviewer 1 Report

After reading the manuscript several times, the focus of the content remains unclear to me. My feeling is that different research approaches were put together to construct a  sound narrative. Unfortunately, I am still confused about the lack of systematic procedure to solve one specific scientific question. Moreover, the results part is difficult to read (mix of introduction, discussion and results) In the following, I summarize my incomplete list of concerns and/or questions:

  • When reading the title of the manuscript the reader expects data derived from stringent cell culture experiments which convincingly show in detail why Calu-3 cells are more susceptible to SARS-Cov-2 infection and replication (the authors use "physiologically relevant cells") than the other cells used. For example, blocking experiments in TMPRSS2 positive cells might have been an interesting aspect to to investigate virus entry mechanisms (but probably this was not the study intention by the authors). TMPRSS2 and its role in SARS-Cov-2 infection was not further explained in the entire manuscript.
  • Why is Chloroquine used as a negative example? I understand that the authors wanted to differentiate between the known compound that was tested in the WHO-recommended trials. What is the exact reason why mefloquine might be superior over CQ/HCQ with specific regard to the chemical structure as stated in the Conclusion. No chemical structure data are provided for either compound. By the way, it ought to be discussed that there were several studies showing that CQ/HCQ e.g. in combination with zinc were highly effective in reducing hospitalization and mortality when applied early (4-5 days) after upcoming of first symptoms. In many trials, CQ/HCQ was applied too late after disease onset and at too high concentrations to elicit the known clinical effects. Is mefloquine also a zinc ionophore like CQ/HCQ? To my knowledge, RDV had no significant and clear clinical benefit to avoid hospitalization and mortality. However, it was generally used in hospitals which means that patients were treated too late because the virus had already reached the lower respiratory tract, elicited pulmonary dysfunction, cytokine storm etc. The timing of treatment is essential but this aspect is not discussed in the manuscript.
  • As mentioned above, I miss the systematic analysis of the differential interactions between the virus and the different cells used. Instead, indirect data and evidences are provided from external databases to underline the clinical relevance of a therapy with mefloquine (now alone or in combination with RDV). Again, what is the intention of the manuscript? Unfortunately, I was not able to understand the role of these data base analyses and the used technical terms which are not sufficiently explained. 
  • The PCR data are not sufficiently convincing to me. The procedure to systematically compare replication rates in the different cell lines is not sufficiently explained. What does it mean when replication is reduced when the virus has been pretreated before incubation with cells? Was this the case in all cell lines tested? The EM pictures are not convincing in this context. What does it mean? A comparison study with EM in different cell lines might be interesting for descriptive analysis. 
  • Why are monocytes tested here? I understand the general rationale but cytokine storm was not the focus of the manuscript...To systematically show the influence of a given drug on the production of proinflammatory cytokines in monocytes more controls ought to be done and results have to be provided.  Influence of the drug with /without viral infection...this is a seperate paper...!
  • What is the PBPK model and what is the sense to simulate the pharmacological characteristics in this context? The tissue-to-plasma expression....? This could be a seperate paper....
  • The statement that early steps of replication cyle were blocked are not experimentally underlined...rather by indirect evidence...however, this was stated in the Abstract.
  • The statement that mefloquine might be beneficial in blocking SARS-Cov-2 in the CNS is not based on data derived from this study but this is an entirely new aspect...which should be investigated in a seperate study...e.g. with cell lines relevant for the CNS.
  • Overall, a specific and clear focus is lacking in this manuscript which is confusing the reader....I suggest that the authors select one of the interesting issues mentioned in this manuscript in order to scientifically study only one aspect in detail with a clear conclusion.

Author Response

Response to Reviewers` Comments

Editor

  1. Agree that more detail on comparison of the chemical structure between CQ/HCQ and mefloquine should be included in the manuscript.

We added a supplementary table (Table S3) comparing the physical-chemical properties of CQ, HCQ and mefloquine. These properties reinforce mefloquine`s ability to be a donor/acceptor of hydrogen bounds and an pH modulator retaining a lipophilic condition in a physiological environment.

We also included some information related to the compounds` chemical structures and properties in the Introduction and added the chemical structures in the Table 1 of the manuscript.

These and other chemical analysis made us include another author in this revision, our group's Chemist.

  1. The reviewer highlights the timing of treatment and presence of zinc with CQ/HCQ treatment is important for their antiviral activity and the authors should discuss this.

This is a very good point that deserves a fully independent investigation in another manuscript, characterizing, for example: i) the synergistic activity between zinc and CQ, HCQ or mefloquine; ii) the antiviral activity of zinc and these molecules administered independently and in salt formulations; iii) the ability of these compounds to control acid-base equilibrium at endosomal level complexed or not with zinc.

Nevertheless, we discussed this point in the revised version of our manuscript. It is possible that mefloquine binds zinc, because according to the Pearson`s acid-base theory mefloquine is a borderline base - which are likely to bridge with cations (II). However, in endosomal acid conditions, it is more likely that the nitrogen at the quinoline moiety will interact more favorably with H+ than zinc, because this nitrogen has a pK lower than 3.0. We added this information at the discussion, citing the new table described above and new references on zinc/CQ anti-SARS-CoV-2 activity.

With respect to timing, reviewer 1 formulates a similar questions taking as reference the time from onset of illness.  We cannot address it in cell culture experiments. The timing to treat the cells is discussed below.

  1. Clarify 'pre-treatment' of cells and virus particles in the materials and methods section.

We performed different types of infection assays. In Table 1 and Figure 1, we infected the cells and treated them 1hour post-infection. In this experiment (described as yield reduction assay in the methodology), viruses were allowed to perform several rounds of replication, because supernatant was harvested 24-48 hours post-infection. These experiments showed us mefloquine`s pharmacological parameters to inhibit virus replication.

In another section of our results, we faced the challenge of addressing the mechanism of action of a drug targeting the host, such as defined in this especial issue of the Viruses journal. To do so, we applied differential expression analysis on SARS-CoV-2 and mefloquine on the host cell. These analysis highlighted us that mefloquine was interfering in the virus entry process. We next confirmed by experimental approaches mefloquine`s ability to reduce virus entry. Only under this circumstance, in which we had to allow only the entry process to occur,  we needed to use pre-treatment. If we treated the cells 1 hour after infection, the ability to monitor the drug`s effect in the entry process would have been jeopardized. We made this approach clearer while describe it in the results. 

  1. Additional data supporting their claims regarding pro-inflammatory cytokines in monocytes is needed. e.g. RT-qPCR and ELISAs data for both IL-6 and TNF-a. Perhaps enrichment plots of pro-inflammatory signalling pathways would help convince the readers/reviewers of mefloquine's activity.

We expanded Figure 1F to the new Figure 2 in the revised manuscript. We added more concentrations of the compounds to evaluate the inhibition of virus replication in monocytes by RT-qPCR, along with quantification of cell death. We also added LPS as a positive control for the cytokine analysis and, to improve data transparency, we presented individual monocytes donors quantifications of IL-6 and TNF-a by ELISA as heatmaps.

  1. Agree that the claim about mefloquine and the CNS is not supported by the study and this should be clarified.

We meant to associate mefloquine`s ability to penetrate CNS with its anti-SARS-CoV-2, but we don’t want to oversell our data. Therefore, we understand the editor/reviewer`s point and removed these associations from the manuscript.

Reviewer #1

  1. After reading the manuscript several times, the focus of the content remains unclear to me. My feeling is that different research approaches were put together to construct a  sound narrative. Unfortunately, I am still confused about the lack of systematic procedure to solve one specific scientific question. Moreover, the results part is difficult to read (mix of introduction, discussion and results) In the following, I summarize my incomplete list of concerns and/or questions:

We disagree with the referee`s view on our manuscript. This investigation has been seriously taken by the authors, from different countries and institutions, and favorably reviewed by the other peers. The rationale for the conduction of our investigation was the following:

  1. We first aimed to evaluate whether mefloquine would inhibit the replication of the novel coronavirus and then determine the drug`s pharmacological parameters in different cell models. Among them, Calu-3 cells, a cell line that recapitulate type II pneumocytes - the epithelial cells preferentially infected by SARS-CoV-2 in the lungs;
  2. We also aimed to evaluate mefloquine`s ability to reduced viral loads, virus-induced cell death and production of inflammatory cytokines in primary monocytes. COVID-19 is characterized by a systemic increase in inflammatory mediators. During the inflammatory phase of the disease, monocytes play a pivotal function in the antiviral host response. However, due to their hyperactivated state and massive cell death caused by SARS-CoV-2 infection of these cell, monocytes also contribute to the release of pro-inflammatory cytokines and exacerbation of lung tissue damage;
  3. Since mefloquine is a drug that targets the host, we next investigated its mechanism of action by studying cellular pathways modulated by mefloquine through different analyses of differential expression. By this same approach, we identified the cell pathways regulated by SARS-CoV-2 infection and then compared mefloquine and SARS-CoV-2 infection expression signatures in order to get insights on how the drug could revert the effects of the disease;
  4. Next, we experimentally confirmed the findings of the differential expression analysis that mefloquine impairs endocytosis-related pathways, by showing that this drug reduces SARS-CoV-2 endocytosis-mediated entry trough infection assays and transmission electron microscopy;
  5. Considering that mefloquine is a host-targeted drug, we explored whether it could be used in combination with direct-acting antiviral drugs. We circumstantially used remdesivir as an example, but mefloquine could also be combined with drugs acting in other stages of the virus replication cycle;
  6. To think about mefloquine and its chemical structure as a candidate to the development of a host-acting drug a that affect SARS-CoV-2 replication, it is crucial to predict whether this drug in vitro pharmacological parameters would be physiologically achievable in vivo. We used physiologically based pharmacokinetic modelling (PBPK) to study mefloquine`s bioavailability and plasma exposure after different dose regimens.

We don’t see that our conduction of this investigation as misguided. 

  1. When reading the title of the manuscript the reader expects data derived from stringent cell culture experiments which convincingly show in detail why Calu-3 cells are more susceptible to SARS-Cov-2 infection and replication (the authors use "physiologically relevant cells") than the other cells used. For example, blocking experiments in TMPRSS2 positive cells might have been an interesting aspect to to investigate virus entry mechanisms (but probably this was not the study intention by the authors). TMPRSS2 and its role in SARS-Cov-2 infection was not further explained in the entire manuscript.

It has already been described and cited in the manuscript  the inability of CQ to inhibit SARS-CoV-2 replication in cells expressing TMPRSS2, such as Calu-3 cells. On the other hand, SARS-CoV-2 replication in Calu-3 cells is susceptible to mefloquine. A specific study related to the participation of the TMPRSS2 engagement in the virus replication cycle may be necessary in the future.

  1. Why is Chloroquine used as a negative example? I understand that the authors wanted to differentiate between the known compound that was tested in the WHO-recommended trials. What is the exact reason why mefloquine might be superior over CQ/HCQ with specific regard to the chemical structure as stated in the Conclusion. No chemical structure data are provided for either compound. By the way, it ought to be discussed that there were several studies showing that CQ/HCQ e.g. in combination with zinc were highly effective in reducing hospitalization and mortality when applied early (4-5 days) after upcoming of first symptoms. In many trials, CQ/HCQ was applied too late after disease onset and at too high concentrations to elicit the known clinical effects. Is mefloquine also a zinc ionophore like CQ/HCQ? To my knowledge, RDV had no significant and clear clinical benefit to avoid hospitalization and mortality. However, it was generally used in hospitals which means that patients were treated too late because the virus had already reached the lower respiratory tract, elicited pulmonary dysfunction, cytokine storm etc. The timing of treatment is essential but this aspect is not discussed in the manuscript.

Please, se responses to Editor`s comments 1 and 2.  We agree that PK/PD relationship for RDV is not very favorable.

  1. As mentioned above, I miss the systematic analysis of the differential interactions between the virus and the different cells used. Instead, indirect data and evidences are provided from external databases to underline the clinical relevance of a therapy with mefloquine (now alone or in combination with RDV). Again, what is the intention of the manuscript? Unfortunately, I was not able to understand the role of these data base analyses and the used technical terms which are not sufficiently explained. 

For better guidance of the experimental rationale, please see the response to your comment 1.

  1. The PCR data are not sufficiently convincing to me. The procedure to systematically compare replication rates in the different cell lines is not sufficiently explained. What does it mean when replication is reduced when the virus has been pretreated before incubation with cells? Was this the case in all cell lines tested? The EM pictures are not convincing in this context. What does it mean? A comparison study with EM in different cell lines might be interesting for descriptive analysis. 

Please, see the response to comment 3 of the Editor.

  1. Why are monocytes tested here? I understand the general rationale but cytokine storm was not the focus of the manuscript...To systematically show the influence of a given drug on the production of proinflammatory cytokines in monocytes more controls ought to be done and results have to be provided.  Influence of the drug with /without viral infection...this is a seperate paper...!

Please, see the response to the Editor`s comment 4 and to your first comment.

  1. What is the PBPK model and what is the sense to simulate the pharmacological characteristics in this context? The tissue-to-plasma expression....? This could be a seperate paper....

Please, see response for the first comment regarding the rationale of our investigation.

  1. The statement that early steps of replication cyle were blocked are not experimentally underlined...rather by indirect evidence...however, this was stated in the Abstract.

To alleviate, we changed this statement in the Abstract and now reads: “Mefloquine reduced SARS-CoV-2 entry…”

  1. The statement that mefloquine might be beneficial in blocking SARS-Cov-2 in the CNS is not based on data derived from this study but this is an entirely new aspect...which should be investigated in a seperate study...e.g. with cell lines relevant for the CNS.

Please see the response to the Editor`s comment 5.

  1. Overall, a specific and clear focus is lacking in this manuscript which is confusing the reader....I suggest that the authors select one of the interesting issues mentioned in this manuscript in order to scientifically study only one aspect in detail with a clear conclusion.

Please see the response to your initial comment for rationale guidance.

Reviewer 2 Report

This is a well written manuscript by Sacramento et al. describing invitro and Human Lung Cancer Cell Line (Calu-3) activity of the antimalarial agent Mefloquine to inhibit SARS-COV-2 infection. Detailed report, with clinical significant amidst the ongoing pandemic. Few points to the authors:

Active clinical trial for Mefloquine efficacy as prophylaxis Against COVID-19 is currently recruiting (NCT04847661) and should be referred to in the manuscript.

It is interesting to see data on the whole-body PBPK model, we know from prior PKPD studies that relationship between EC90 (in vivo) and EC90 (in vitro) for mefloquine is complex. The in vivo concentration-effect curve is shifted to the right of the in vitro concentration-effect curve because of binding to proteins and other factors. The level of protein binding is generally considered to be very high, approximately 98% for mefloquine as mentioned in the manuscript. Discussion about this was added to discussion section as limitation, this is considered major limitation of clinical application. Variability in blood level related to many factors has been well described in literature.

Repeated emphasis in the paper about CNS penetration is unexplained to me, as CNS infection with SARS-COV-2 is not fully understood and this hitherto remains area of debate. Recommend removing this or modify as this is clinically not proven yet.

Author Response

  1. Active clinical trial for Mefloquine efficacy as prophylaxis Against COVID-19 is currently recruiting (NCT04847661) and should be referred to in the manuscript.

We included this information in the Introduction and Discussion sections.

  1. It is interesting to see data on the whole-body PBPK model, we know from prior PKPD studies that relationship between EC90(in vivo) and EC90 (in vitro) for mefloquine is complex. The in vivo concentration-effect curve is shifted to the right of the in vitro concentration-effect curve because of binding to proteins and other factors. The level of protein binding is generally considered to be very high, approximately 98% for mefloquine as mentioned in the manuscript. Discussion about this was added to discussion section as limitation, this is considered major limitation of clinical application. Variability in blood level related to many factors has been well described in literature.

Indeed, drug-protein binding is an important feature to be considered during the study of novel/repurposed molecules. For COVID-19 treatments, the comparison between in vitro-derived activities to total plasma concentrations has been discussed since only the unbound drug fraction is assumed to be able to exert antiviral activity. According to a recent consensus on this subject (PMID: 33113246), drug binding in vitro also occurs due to bind to culture plastics and culture media constituents`, mainly fetal bovine/human serum.  Previous studies with lopinavir showed that a culture medium containing 5-10% of serum bound 93-961% of the drug and had the same amount of free drug in vivo, making comparable lopinavir`s in vitro EC90 and in vivo Cmax. In our experimental infections assays with mefloquine, we used 10% of fetal bovine serum in Calu-3 cells or 5% of human serum in monocytes. We added this information in the discussion to alliviate the readers' concern. Similarly, we added that direct conversion from in vitro pharmacological parameters to in vivo concentrations may have limitationzls.

  1. Repeated emphasis in the paper about CNS penetration is unexplained to me, as CNS infection with SARS-COV-2 is not fully understood and this hitherto remains area of debate. Recommend removing this or modify as this is clinically not proven yet.

We removed these associations with CNS from the manuscript.

Reviewer 3 Report

The authors aimed to investigate anti-SARS-CoV-2 mechanism of action of mefloquine in cells relevant for the physiopathology of COVID- 19.

The study covers some issues that have been overlooked in other similar topics. The structure of the manuscript appears adequate and well divided in the sections. Moreover, the study is easy to follow, but some issues should be improved. The manuscript needs moderate grammar correction. Please also check typos thorough the text.

Introduction section: Will be useful to the reader to add some interesting recent literature about the updates against molecular mechanisms related to  SARS-CoV-2 outbreak and related tools to counteract the same (please see and briefly discuss:  PMID: 33024749).

Conclusion Section: This paragraph is missing. Please add it.

Author Response

  1. Introduction section: Will be useful to the reader to add some interesting recent literature about the updates against molecular mechanisms related to  SARS-CoV-2 outbreak and related tools to counteract the same (please see and briefly discuss:  PMID: 33024749).

We included some information in the Introduction and discussion sections and cited the suggested reference.

  1. Conclusion Section: This paragraph is missing. Please add it.

We included a Conclusion section in the revised manuscript.

Round 2

Reviewer 1 Report

General: The manuscript has been improved. Although I still feel that the content is pretty overloaded. There are many typos. Language has to be checked.

Specific: The electron microscopy data are from n=1 and are not convincing. The new citation #62 should be substituted by the publication of real world data by Derwand R et al. COVID-19 outpatients: early risk-stratified treatment with zinc plus low-dose hydroxychloroquine and azithromycin: a retrospective case series study. Int J Antimicrob Agents. 2020 Dec;56(6):106214. doi: 10.1016/j.ijantimicag.2020.106214. Epub 2020 Oct 26. PMID: 33122096; PMCID: PMC7587171.

Author Response

  1. The manuscript has been improved. Although I still feel that the content is pretty overloaded. There are many typos. Language has to be checked.

The text was revised and language improvements were made.

  1. The electron microscopy data are from n=1 and are not convincing.

The electron microscopy analysis was performed in four independent experiments. Several micrographs were obtained from each experiment and representative pictures were included as Figure 4 B and C, along with the quantification of the number of virions in endosomes from all the micrographs of the four experiments (Figure 4D). Anyway, the Reviewer remark highlighted us to add the statistical analysis in Figure 4A and D.

To make it clearer, we added this information in the legend of Figure 4.

  1. The new citation #62 should be substituted by the publication of real world data by Derwand R et al. COVID-19 outpatients: early risk-stratified treatment with zinc plus low-dose hydroxychloroquine and azithromycin: a retrospective case series study. Int J Antimicrob Agents. 2020 Dec;56(6):106214. doi: 10.1016/j.ijantimicag.2020.106214. Epub 2020 Oct 26. PMID: 33122096; PMCID: PMC7587171.

The citation was substituted according to the reviewer`s suggestion.

Reviewer 2 Report

Review points covered 

Author Response

We are happy that the Reviewer is satisfied with the current version.